# Gravitational-wave localization alone can probe origin of stellar-mass black hole mergers

I. Bartos[1,4], Z. Haiman[2,3], Z. Marka[1], B.D. Metzger[1], N.C. Stone[1] & S. Marka[1]

The recent discovery of gravitational waves from stellar-mass binary black hole mergers by the Laser Interferometer Gravitational-wave Observatory opened the door to alternative probes of stellar and galactic evolution, cosmology and fundamental physics. Probing the origin of binary black hole mergers will be difficult due to the expected lack of electromagnetic emission and limited localization accuracy. Associations with rare host galaxy types—such as active galactic nuclei—can nevertheless be identified statistically through spatial correlation. Here we establish the feasibility of statistically proving the connection between binary black hole mergers and active galactic nuclei as hosts, even if only a sub-population of mergers originate from active galactic nuclei. Our results are the demonstration that the limited localization of gravitational waves, previously written off as not useful to distinguish progenitor channels, can in fact contribute key information, broadening the range of astrophysical questions probed by binary black hole observations.

[1] Columbia Astrophysics Laboratory, 550 West 120th Street, New York, NY 10027, USA. [2] Department of Astronomy, Columbia University, 550 West 120th Street, New York, NY 10027, USA. [3] Department of Physics, New York University, 726 Broadway, New York, NY 10003, USA. [4] Present address: Department of Physics, University of Florida, PO Box 118440, Gainesville, FL 32611-8440, USA. Correspondence and requests for materials should be addressed to I.B. (email: ibartos@phys.columbia.edu)

Stellar-mass binary black hole (BBH) mergers are the most promising gravitational-wave (GW) sources for earth-based detectors such as the Laser Interferometer Gravitational-wave Observatory (LIGO)[1]. As LIGO's sensitivity will gradually increase in the coming years, and other GW facilities come online[2–4], the rate of GW observations will increase to ~1 per day[5, 6], making it possible to comprehensively study GW source populations.

Binary black holes (BBHs) can be formed either from massive stellar binaries[7] or via dynamical interactions in dense stellar systems, including globular clusters and galactic nuclei[8]. Identifying the BBH formation channel will be a key step in using BBH mergers as cosmic probes. However, this identification for a single merger is difficult without an electromagnetic counterpart[5], which can only be produced if the binary is surrounded by sufficient gas that can be accreted. This is not expected for most formation channels (but see refs. [9–13]).

Possible observational clues to the binary's origin include the binary mass ratio[5], mass distribution[14], black hole spin[15, 16], and orbital eccentricity close to merger[17]. However, the efficiency of these clues in differentiating between formation channels is uncertain, and is dependent on complex stellar evolution and dynamics processes.

Accurate GW localization, and the identification of the host galaxy, could be an additional important clue in constraining the formation channel. However, the large number of possible host galaxies will require highly accurate localization that may only be achieved for a small fraction of GW observations[18], or with future GW detector networks. The situation is complicated further if multiple formation channels are present, ultimately requiring the statistical study of rare events.

Here we investigate the prospects of statistically proving the connection of BBH mergers with rare hosts, focusing on luminous active galactic nuclei (AGN). Copious gas inflow to the nuclei of bright AGN (i.e. quasars) provides a potential site for finding massive black holes[19] that can enhance the merger rate as BBHs embedded in their accretion disks rapidly merge due to gas dynamical friction[9, 10]. AGN represent a small fraction (~1%) of galaxies, making their identification as hosts feasible. The results we present in the following can be extended and applied to other similarly rare host populations.

## Results

**Number of detections needed to establish correlation.** We aim to take advantage of spatial correlation between the distribution of a host population, namely AGN, and the location of origin of detected GWs. The origin of each detected GW can be localized to within a finite volume at high confidence[20]. GWs originating from an AGN population will preferentially come from regions in the universe with higher AGN number density, while GWs of other origin will show no such preference.

The host population is assumed to be a known set of point sources. Since Advanced LIGO-Virgo will only be sensitive to mergers at redshift $z \lesssim 1$[21], it is possible to achieve a sufficiently complete AGN catalog within this range. Additionally, even the closest AGN have much smaller angular diameter than the precision of GW reconstruction, hence we can treat AGN as point sources. We will further assume that AGN are randomly, uniformly distributed within the local universe, a conservative assumption given that they are known to cluster[22]. We will also assume that AGN are spatially uncorrelated with alternative merger sources (see further discussion of this point below).

A network of GW detectors can constrain the location of origin of a GW, generating a 3D probability density. This probability distribution is typically expressed as a 3D localization comoving volume (hereafter localization volume), at a threshold confidence or credible level $CL_{sky}$, often chosen to be 90%. For simplicity, we will use this localization volume at $CL_{sky} = 90\%$, without taking advantage of the inner probability density.

We calculated the minimum number of BBH GW observations, denoted with $N_{gw,3\sigma}$, that is necessary to establish AGN as the host of a some of the detected BBHs, at a median $3\sigma$ significance (see Methods section). The number $N_{gw,3\sigma}$ depends on the fraction $f_{agn}$ of observed GWs originating from AGN. We used a Monte Carlo analysis described in the Methods section to obtain $N_{gw,3\sigma}$ as a function of $f_{agn}$. We found the scaling relation

$$N_{gw,3\sigma} \propto f_{agn}^{-2}. \qquad (1)$$

This can be intuitively expected: the s.d. of the total number of AGN for all GW detections together scales with $N_{gw,3\sigma}^{1/2}$, while the number of "signal" AGN scales with $N_{gw,3\sigma} f_{agn}$, corresponding to

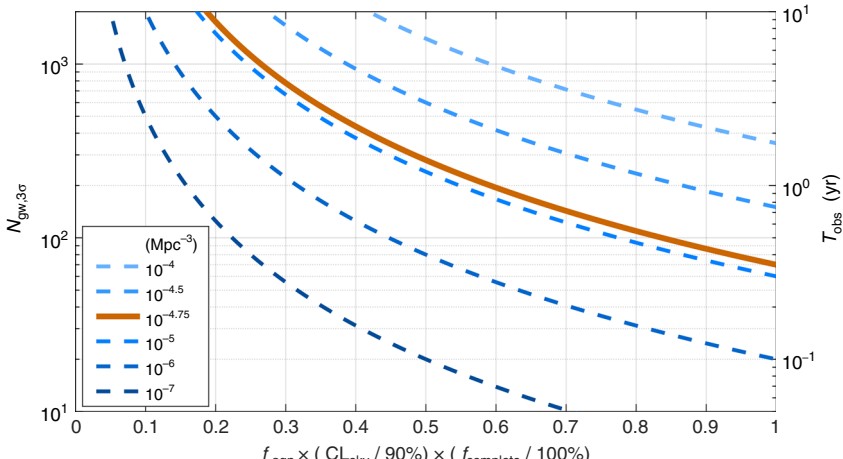

**Fig. 1** Number of gravitational-wave detection required for host identification. Number of detections ($N_{gw,3\sigma}$) needed to establish active galactic nuclei (AGN) as host population at a median $3\sigma$ significance, as a function of the fraction ($f_{agn}$) of gravitational-wave (GW) detections originating from AGN. The results are shown for different assumed AGN number densities (see legend), with fiducial density $\rho_{agn} = 10^{-4.75}$ Mpc$^{-3}$ [23, 24], for the Advanced Laser Interferometer Gravitational-wave Observatory (LIGO)-Virgo network at design sensitivity. On the right hand side axis, we mark the necessary observation duration corresponding to numbers of detection, assuming 200 detections per year. We also indicated in the label of the x axis the results' scaling with the sky volume credible level used ($CL_{sky}$) and host catalog completeness ($f_{complete}$)

a signal-to-noise ratio SNR $\propto N_{\mathrm{gw},3\sigma}^{1/2} f_{\mathrm{agn}}$. Interpreting our detection threshold as a fixed SNR, we get back Eq. (1).

Our results for $N_{\mathrm{gw},3\sigma}$ are shown in Fig. 1. Here, we applied Monte Carlo analysis to evaluate the case $f_{\mathrm{agn}} = 1$, and used Eq. (1) to show scaling with $f_{\mathrm{agn}}$.

We take a fiducial AGN density $\rho_{\mathrm{agn}} = 10^{-4.75}\,\mathrm{Mpc}^{-3}$, corresponding to AGN with Eddington ratios greater than 1%[23, 24]. If AGN-induced mergers are concentrated among the highest Eddington ratio systems, this density will be lower (by, e.g., one order of magnitude if the Eddington ratio must be above 30%). In our fiducial scenario, we find that ~70 detections would be sufficient to statistically prove the BBH-AGN connection if all BBH mergers occurred in AGN. For the expected BBH merger rate of ~60 $\mathrm{Gpc}^{-3}$ per year[5], this corresponds to a few months of LIGO-Virgo[25] observation time at design sensitivity, indicating that this scenario would yield results quickly (possibly even before LIGO and Virgo reach design sensitivity). With the current uncertainty in BBH merger rate of 9–240 $\mathrm{Gpc}^{-3}$ per year[5], the required time is within a month and a few years. Even if only a fraction of mergers occur in AGN, we find that a 5-year observation period is likely sufficient to statistically prove the BBH-AGN connection if the AGN fraction is at least 25%.

**Dependence on the number density of host galaxies**. To understand our results' dependence on the uncertain AGN number density, and to demonstrate the method's applicability to other rare host types, we also obtained results for a range of different $\rho_{\mathrm{agn}}$ number densities. While a sparser source population leads to even less detections needed for the identification of a host population, we found that, even for number densities of $10^{-4}\,\mathrm{Mpc}^{-3}$, a 5-year observation period will establish the BBH-AGN connection for $f_{\mathrm{agn}} \gtrsim 0.5$. With higher host number densities it becomes difficult to prove an AGN connection solely using GW localizations, although the construction of additional GW detectors (Kamioka Gravitational Wave Detector, or KAGRA[26] and LIGO India[4]), and potentially the inclusion of marginally significant GW events, will further improve the situation.

**Importance of the best localized mergers**. To further investigate how GWs with different localization volumes contribute to the results, we carried out the Monte Carlo analysis described above, but with fixed localization volume size. This analysis confirmed that, for fixed localization volume, our method described above is equivalent to combining AGN from all GW detections, and looking for a $3\sigma$ deviation in the overall number. The GW detection number threshold in this case, for $N_{\mathrm{gw},3\sigma} \gg 1$, is $N_{\mathrm{gw},3\sigma} = 9\rho_{\mathrm{agn}}V$, where $V$ is the fixed localization volume.

To determine whether our statistical analysis mainly relies on a few well-localized GWs or if less-well-localized mergers are also useful, we reran our Monte Carlo study for $\rho_{\mathrm{agn}} = 10^{-4.75}\,\mathrm{Mpc}^{-3}$ and $f_{\mathrm{agn}} = 1$, with the modification that the analysis only used GW localization volumes below a cutoff volume $V_{\mathrm{cutoff}}$. We measured $N_{\mathrm{gw},3\sigma}$ as a function of the fraction $f_V(V_{\mathrm{cutoff}})$ of localization volumes below $V_{\mathrm{cutoff}}$. We found that $V_{\mathrm{cutoff}} \gtrsim 10^5\,\mathrm{Mpc}^3$ did not meaningfully change $N_{\mathrm{gw},3\sigma}$, but for lower $V_{\mathrm{cutoff}}$, we found a quick deterioration. We conclude that those events drive our constraints in whose localization volumes we expect $\lesssim 2$ interloper quasars on average. For $\rho_{\mathrm{agn}} = 10^{-4.75}\,\mathrm{Mpc}^{-3}$, this is the top ~5% best localized events.

**Discussion**
Our conclusions are the following: firstly, for our fiducial number density $\rho_{\mathrm{agn}} = 10^{-4.75}\,\mathrm{Mpc}^{-3}$ and $f_{\mathrm{agn}} = 1$, ~70 observations will be sufficient, on average, to statistically establish the BBH-AGN connection at $3\sigma$ significance. With an expected rate of BBH merger detection of 200 $\mathrm{yr}^{-1}$, this corresponds to 4 months of observation time. Secondly, we quantified the efficiency of establishing BBH-AGN for fractional AGN contributions (Fig. 1). For a fractional contribution $f_{\mathrm{agn}}$, the number of detections sufficient on average to establish the BBH-AGN connection is $N_{\mathrm{gw},3\sigma} = 70 f_{\mathrm{agn}}^{-2}$. Finally, incorporating the ~5% best localized BBH mergers is sufficient to produce close to optimal constraints.

Our results demonstrate that correlation between the location of AGN and GWs alone can be used to establish the BBH-AGN connection with a few years of observation with Advanced LIGO-Virgo at design sensitivity, making this a competitive approach compared to more model dependent probes that use reconstructed BBH properties, or uncertain electromagnetic counterparts.

In principle, LIGO events can be spatially correlated with AGN even if they are unrelated to AGN, but occur in galaxies whose spatial distribution is correlated with AGN. The cross-correlation length between local galaxies and quasars is ~6 Mpc[27], which is an order of magnitude smaller than the linear size of the typical LIGO error volume[18], so we expect this effect to be small, unless the events occur in rare galaxy sub-types that have a stronger correlation with AGN.

To confirm that galaxy-AGN correlations do not affect our results, we repeated our calculations with the addition of the galaxy-AGN cross-correlation function $\xi(r) = (r/6.6h^{-1}\,\mathrm{Mpc})^{-1.69}$ from Shen et al.[27] in calculating the expected number of AGN within the GW localization volume. We found negligible difference in our results compared to our baseline calculations with no assumed correlations.

The technique and results presented here can also be applied to other rare host populations. For instance, a relevant channel is BBHs formed by GW capture in the dense stellar-mass black hole populations of galactic nuclei[17]. This scenario preferentially occurs in the densest nuclei, and therefore could be strongly correlated with E + A galaxies. E + As are post-starburst galaxies that represent ≈0.2% of all $z \approx 0$ galaxies but host an order unity fraction of stellar tidal disruption events[28, 29]; preliminary evidence suggests that this is due to overdense central star clusters[30].

There are important next steps that will further enhance the prospects of our analysis. AGN are relatively strongly clustered, which can enhance the signal to noise ratio over a random AGN distribution assumed here. Different host populations can be spatially correlated, which needs to be taken into account before the BBHs can be inferred to reside in AGN (rather than just correlated with them statistically; see, e.g., ref. [31]). Strong correlation between two host types would render the present method ineffective in differentiating between them. Combining our error region analysis with other observables (binary mass, spins, etc.) and host galaxy properties that correlate with binary rate will significantly enhance search sensitivity. If suitable models are available, reconstructed BBH properties can add to the differentiating power of a search. If an AGN-BBH connection is established, a modified version of the present method can be used to determine the specific fraction of BBH mergers originating from AGN. This method requires that we either have a complete AGN catalog in the BBH merger localization volumes, or at least we know the level of completeness $f_{\mathrm{complete}}$.

We further emphasize the need for detailed AGN catalogs out to redshift of $z \sim 0.2$; these catalogs will provide the backbone of any statistical BBH-AGN connection[32, 33]. In principle, the nearby AGN which host most LIGO events, with $M_{\mathrm{bh}} \gtrsim 10^6\,M_\odot$) should be bright enough to be detectable in large all-sky surveys[34]. In practice, however, existing spectroscopically confirmed quasar catalogs appear incomplete. In the full

comoving volume out to $z = 0.1$ (~0.3 Gpc$^3$) we expect to find ~10,000 AGN. This number is based on extrapolating spectroscopic measurements for faint nearby AGN to the rest of the sky. On the other hand, the latest quasar catalog from the Sload Digital Sky Survey (SDSS), covering $\approx$10,000 deg$^2$ [35] contains only 232 quasars at $z < 0.1$, implying that it is only 10% complete. Deeper spectroscopic surveys exist but target only a small fraction of the sky. More targeted cataloging efforts are needed to take full advantage of GW observations. Fortunately, galaxy catalogs are already in focus for multimessenger GW observations[36], and future surveys, such as the Large Synoptic Survey Telescope (LSST), will significantly accelerate these efforts[37]. Additionally, with no full-sky catalogs, it is also feasible to identify galaxies in follow-up surveys of individual GW localization volumes[33, 38], provided that a statistically sufficient number of random fields outside the GW localization volumes are also surveyed to similar depths to serve as a control sample.

## Methods

**Statistical analysis of host population.** Let the localization volume of the $i^{th}$ detected GW be $V_i$. For a GW not originating from an AGN, the number of AGN within $V_i$ will follow a Poisson distribution with mean $\lambda_i = \rho_{agn} V_i$, where $\rho_{agn}$ is the number density of AGN. The probability of having $N_{agn,i}$ AGN's within $V_i$ for our null hypothesis is

$$B_i\left(N_{agn,i}\right) = \text{Poiss}\left(N_{agn,i}, \rho_{agn} V_i\right). \qquad (2)$$

If the GW originated in an AGN, then there will be 1 guaranteed AGN, with the number of additional AGN following a Poisson distribution with $\lambda_i$ mean. The probability of having $N_{agn,i}$ AGN in $V_i$ volume for our alternative hypothesis is therefore

$$S_i\left(N_{agn,i}\right) = \text{Poiss}\left(N_{agn,i} - 1, \rho_{agn} V_i\right). \qquad (3)$$

While a single GW detection may not be sufficient to determine its host population of origin, combining multiple GW observations will increase the likelihood that we can statistically prove the connection between BBH mergers and AGN. For the alternative hypothesis that a fraction $f_{agn}$ of the detected GWs originated from AGN and $(1 - f_{agn})$ from some other host population, we obtain the likelihood

$$L\left(f_{agn}\right) = \prod_i \left[f_{agn} S_i + \left(1 - f_{agn}\right) B_i\right] \qquad (4)$$

where the product is over all detected BBH mergers during the observation period.

While we do not know $f_{agn}$ a priori, here we will use a specific value in the likelihood ratio test. For estimating $f_{agn}$, one can maximize $L(f_{agn})$ with respect to $f_{agn}$. Since we focus here on sensitivity and not the precision of parameter reconstruction, we will ignore this step.

The test statistic of a set of detected GWs will be the likelihood ratio

$$\lambda = 2 \log\left[\frac{L\left(f_{agn}\right)}{L(0)}\right] \qquad (5)$$

The significance of an ensemble of GW observations will be determined by comparing the observed $\lambda$ value to $\lambda$'s background distribution $P_{bg}(\lambda)$. We will reject the null hypothesis in favor of $f_{agn}$ fraction of the GWs originating from AGN if the GWs' $\lambda$ corresponds to a $p$-value less than $3\sigma$ (=0.00135).

We now determine how many GW detections will be sufficient to reject the null hypothesis with $3\sigma$ significance, given an $f_{agn}$ fraction. We characterize this number, $N_{gw,3\sigma}(f_{agn})$, by the median number of detections that is sufficient to reject the null hypothesis at $3\sigma$ level.

**Monte Carlo simulations.** To find $N_{gw,3\sigma}(f_{agn})$, we use Monte Carlo simulations (see ref. [39]). We adopt the distribution of GW localization volumes obtained by Chen & Holz at 90% confidence level[18] for the LIGO-Virgo GW detector network at design sensitivity. We take their results for $10\,M_{\odot} - 10\,M_{\odot}$ BBH mergers as a characteristic binary. As only 90% of GWs will be located within these localizations volumes, this decreases the effective AGN fraction $f_{agn}$ by a factor of 0.9, which we take into account in the following calculations.

For one Monte Carlo realization, we assume a GW detection number $N_{gw}$, and (effective) AGN fraction $f_{agn}$. For each GW detection, we randomly assign an AGN origin with $0.9 f_{agn}$ probability, otherwise it is considered to originate from another host type. For GW detection $i$, we assign a random localization volume $V_i$ drawn from the Chen-Holz distribution described above, and generate a random AGN number within this volume, using the distributions in Eqs. (2) and (3). We then calculate $\lambda$ corresponding to this realization using Eq. (5). We similarly obtain

background realizations using Eq. (2), and use them to determine the p-values corresponding to the realizations with GWs from AGN.

We repeat the above Monte Carlo analysis for a range of $N_{gw}$ values for a given $f_{agn}$ in order to determine the threshold number $N_{gw,3\sigma}(f_{agn})$.

**Data availability.** The data that support the findings of this study are available from the corresponding author upon request.

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

## Acknowledgements

We thank Jules Halpern for useful discussions. I.B., Z.M., and S.M. are thankful for the generous support of Columbia University in the City of New York and of NSF grant PHY-1404462. Z.H. acknowledges support from NASA grant NNX15AB19G and a Simons Fellowship for Theoretical Physics. B.D.M. acknowledges support from NASA grant NNX16AB30G and the Research Corporation for Science Advancement Scialog Program grant RCSA23810. NCS acknowledges support by NASA through Einstein Postdoctoral Fellowship Award Number PF5-160145.

## Author contributions

All authors contributed to the origination of the idea for the project and worked out collaboratively the general details. I.B. performed the numerical simulations.

## Additional information

**Competing interests:** The authors declare no competing financial interests.

