## [Peer Review File · Nature Communications]

Reviewer #1 (Remarks to the Author):

The Manuscript "Gravitational-Wave Localization Alone Can Probe Origin of Stellar-Mass Black Hole Mergers" describes a very simple and neat idea to test possible formation scenarios of the black hole binaries (BHBs) observed by LIGO. The errorboxes of GW observations can be correlated with putative sub-populations of objects that can be candidate hosts of the events in a statistical way. This approach works if a specific model predict BHBs would be formed in particularly rare environments. This is the case for the AGN disk formation scenario. Although one may argue there are more conventional and natural scenarios to form these objects, the technique is nonetheless interesting, if anything to rule out this specific scenario. Most importantly, it shows a novel and clever way to try use limited information from poor sky localization to address an interesting astrophysical problem.

The paper is clearly written and straightforward. The thing I am most concerned with is the level of simplification in the calculation. For example, one of the implicit assumptions is that every event that is generated with an AGN has an associated AGN observed in the errorbox (except for the 0.9 correction due to the assumed 90% confidence errorbox). However, this does not necessarily have to be the case, as the Authors also acknowledge towards the end of the discussion. I would argue that incompleteness of AGN surveys is a major concern.

In general, lower mass MBHs (say MW type or lower) are hosted in denser environment, therefore one might expect many BHBs formed through the AGN channel to be indeed hosted by galaxies with MBHs $\sim 10^6 M_{\text{sun}}$ or less, possibly accreting at 10^{-2} Eddington (fig 11 of Greene & Ho 2007, for example). How demanding would be to obtain a complete sample of AGNs down to the required sensitivity level for all LIGO sources? Another problem is that the vast majority of low luminosity AGN are likely obscured (e.g. Brusa et al. 2010), therefore one would need a deep X-ray coverage of all LIGO errorboxes to faint X-ray fluxes, would that be feasible?

In light of these concerns I solicit the Authors to:

1-comment on the above and provide relevant requirements for Optical and/or X-ray complete identification of AGNs

2-study the feasibility of the study accounting for possible incompleteness of Quasar catalogs. For example, how does the performance of the method scales if a fraction x of the BHBs are in fact hosted by quasars, but those are not identified in companion surveys?

I think addressing point 2 is essential for drawing sensible astrophysical conclusions. If a fraction x of AGNs are missing (because obscured or below the threshold of optical surveys) and a fraction y of the LIGO BHB comes from AGN, what would this technique tell you? I would assume there might be degeneracies between the 2 parameters (x and y), can that be confidently broken?

Apart from this, I have a couple of minor comments that I'd like the Author to address.

a-After the introduction, the Authors present the results right away. I am not sure this is required in the Nature Communications format. In any case, a bit more of background should be given before this discussion. For example, neither N_{gw} nor f_{agn} are defined at this point, so that equation 1 is essentially meaningless at first glance (well, one can guess, of course). Please define the quantities and spend few lines to frame the results.

b-When talking about the applicability of this technique to other formation channels, the Authors write:

"The number of globular clusters within a galaxy strongly correlates with the mass of the central supermassive black hole, so relatively rare, large black holes near the turnover in the Schechter function are the dominant contributors."

One may argue that this correlation is just a mirror of the SMBH-bulge mass relation and of the fact that the number of globular clusters in spheroidal galaxies is proportional (quite unsurprisingly) to the mass of the spheroid. I do not see how this is different than claiming that the BHBs are most likely originating from galaxies with more stars. Unless the Authors can demonstrate this is not the case, this sentence should be removed.

Reviewer #2 (Remarks to the Author):

In this work, the authors have demonstrated there can be a spatial correlation between the AGNs and the gravitational waves emitted from stellar-mass black hole mergers, by using the test of the statistical hypothesis with Monte Carlo simulations. Nobody has yet known exactly where the black hole merger occurs. It is therefore important to identify its location.

The authors derived the condition that the black hole merger happens at the AGNs at a statistically significant level. Even if the future observations would prove it, the statistical analysis cannot say exactly where the merger occurs. Also why the merger prefers to happen at AGNs still remains mystery. The key to solve these questions is the observations of the electromagnetic counterparts. Through them, one can know where and how the binary stellar-mass black holes are formed, and evolve towards the coalescence. That is why this work should not be so impressive for the general readers.

It would be better for the readers to make the following clearer; the concrete values of the significant level and the significance probability, and their derivation basis. This is because whether the null hypothesis can be rejected clearly depends on their values. If it cannot be rejected, the authors cannot say anything conclusive statistically.

Nevertheless, the statistical analysis that authors gave here is very interesting to the GW scientists and the related researchers. It should be important especially for the future GW observation plan. I would like to suggest the authors to publish this paper from ApJ or MNRAS or the other more appropriate journal, not Nature Communication.

Reviewer #3 (Remarks to the Author):

The authors make an attempt to justify that association of gravitational wave sources with rare host galaxies may be identified statistically.

It is interesting that a very similar case was discussed in paper by Burenin et al. (<http://adsabs.harvard.edu/abs/1998AstL...24..427B>, arXiv:astro-ph/9804274). In this work the authors discovered the association of gamma-ray bursts with AGNs with almost exactly the same method, which is developed in the paper by Bartos et al. Note, that the number of objects considered and the typical size of their localizations in the sky are also very similar.

The association of bright GRBs with AGNs at $z \sim 0.3$ was found at more than 3 sigma confidence by Burenin et al., but as we know today GRB are not associated with AGN directly. The reason of observed association is that GRB hosts follow the same large scale structure as other objects at the

same distances and the correlation length (~ 6 Mpc, i.e. 0.4 deg at $z \sim 0.3$) is in fact of the same order as the size of GRB localization region (1-2 deg). Therefore, the result is correct, i.e. that was

indeed one of the first confirmation that the GRB hosts are located at cosmological distances. However, I think that this example shows very clearly that even if the association between the populations is found with this method, that does not necessarily mean that the populations are directly related.

In the case of LIGO/Virgo GW events, the localization area of well localized ones should be < 10 square degrees (Chen & Holz, 2016), i.e. their typical sizes should be smaller than ~ 3 deg. It is larger, but still of order of the correlation angular scale of

galaxies at $z \sim 0.1-0.3$ (0.3--1 deg). That is why I do not agree with the assumption made by authors at page 6, that the effect of cross-correlation of galaxies with quasars at this angular scale should be small. This is one of the main assumption of the method, so the results may change very significantly if it would be relaxed.

I think that the paper should be modified taking in account the considerations given above.

Reviewer #1 (Remarks to the Author):

COMMENT: The paper is clearly written and straightforward. The thing I am most concerned with is the level of simplification in the calculation. For example, one of the implicit assumptions is that every event that is generated with an AGN has an associated AGN observed in the errorbox (except for the 0.9 correction due to the assumed 90% confidence errorbox). However, this does not necessarily have to be the case, as the Authors also acknowledge towards the end of the discussion. I would argue that incompleteness of AGN surveys is a major concern.

RESPONSE: We agree with the Referee that a sufficiently complete AGN catalog is imperative for the success of the proposed analysis. There are two important points here. (1) As we discuss in the last paragraph before methods, current host catalogs, in particular AGN catalogs, are indeed far from complete. There is, however, significant effort under way to catalog the local universe for GW follow-up surveys. Near future catalogs, for example LSST's photometric redshift catalog, will reach out to the required distances, although for LSST redshift estimates from photometric data require further targeted follow-up to reach sufficient accuracy for the purposes of this analysis. One of our intentions with the present paper is to advocate for relevant galaxy surveys. (2) A possibly even more promising direction is to not wait for large-scale host surveys, but to directly search for hosts, e.g. AGN, in the error region of the best localized GWs. We outlined a possible strategy to accomplish this earlier in [Bartos+1410.0677]. To roughly estimate the required coverage, consider the following. The best localized ~5% of events will be localized to ~10 deg², and will occur at luminosity distances ~500 Mpc [Chen & Holz 0805.2366]. Considering ~100 required total number of GWs to be conclusive, this means that one needs to survey $100 \times 0.05 \times 10 = 50$ deg² of the sky with a depth to be sufficiently complete out to ~500 Mpc. This should be achievable even with limited observational resources.

To make the role of catalog completeness clearer, we modified the label for the x axis of Figure 1 to include a scaling term that shows how the results change if we work with an incomplete catalog. The effect is the same as if the fraction of the BBH mergers that come from uncatalogued hosts were not coming from the channel of interest. We further added to and clarified the text in the last paragraph before the Methods section to point out that there are ongoing efforts in host galaxy cataloging for GW surveys, and to further emphasize the role of "follow-up cataloging."

COMMENT: In general, lower mass MBHs (say MW type or lower) are hosted in denser environment, therefore one might expect many BHBs formed through the AGN channel to be indeed hosted by galaxies with MBHs $\sim 10^6 M_{\text{sun}}$ or less, possibly accreting at 10^{-2} Eddington (fig 11 of Greene & Ho 2007, for example). How demanding would be to obtain a complete sample of AGNs down to the required sensitivity level for all LIGO sources? Another problem is that the vast majority of low luminosity AGN are likely obscured (e.g. Brusa et al. 2010), therefore one would need a deep X-ray coverage of all LIGO errorboxes to faint X-ray fluxes, would that be feasible?

1-comment on the above and provide relevant requirements for Optical and/or X-ray complete identification of AGNs

RESPONSE: To estimate the requirements for a complete catalog, for concreteness, let us consider a SMBH with a mass of $10^6 M_{\text{sun}}$ at a distance of 0.7 Gpc, shining at a bolometric luminosity of $L/L_{\text{Edd}}=f_{\text{Edd}}=0.1$. Further assuming a 10% bolometric correction, this source would appear as a 19.5 mag point source in the optical bands. For comparison, LSST has a field of view of 10 deg^2 , and, in the r-band, a single-visit point-source sensitivity of 24.5 mag. The single-visit exposure time is 30 seconds (ref: LSST science book v2.0; arXiv:0912.0201}). We conclude that all AGN could be easily cataloged in a photometric follow-up campaign with LSST, even if pre-existing data was unavailable.

In fact, AGN with $M_{\text{bh}}=10^6-7 M_{\text{sun}}$, out to a distance of ~ 700 Mpc, are bright enough that they should be in the SDSS spectroscopic database, over the SDSS footprint. Reines et al. 2013 ApJ, 775, 116, performed a systematic search for AGN in low-mass galaxies in the SDSS (DR8) spectroscopic catalog. Their search was limited to $z < 0.055$, for which a re-analysis of the SDSS imaging and spectroscopic data was available from the NASA-Sloan Atlas (NSA). Reines et al. state, at the end of the last paragraph of their Section 3, that a $10^5 M_{\text{sun}}$ BH needs to be radiating at $f_{\text{Edd}}=0.08$ to produce a detectable broad H α line (a key AGN diagnostic) at $z=0.055$. Down to their sensitivity limit, they uncovered 136 AGN, which is approximately 1/3rd of the number (400) of low-mass AGN we expect in their search area ($z < 0.055$, $10,000 \text{ deg}^2$), based on the BH mass function of accreting $\sim 10^6-7 M_{\text{sun}}$ BHs in Greene & Ho (2007). The majority of the events should occur in disks around $> 10^6 M_{\text{sun}}$ BHs (which contain most of the BH mass; see the mass functions in Greene & Ho), and should produce a detectable H α line out to > 3 times large distance, i.e. to at least 700 Mpc.

In summary: we believe the required optical AGN catalog currently does not exist. However, the relevant AGN would be bright enough to be in the SDSS spectroscopic database. The analysis required to look for these AGN has not been performed beyond $z=0.055$. The completeness of any catalog created through this analysis is unfortunately difficult to estimate, because: (i) not all galaxies were targeted in SDSS for spectroscopy to begin with, and (ii) AGN can be missed due to starlight contamination from the host, and (iii) line-ratio diagnostics can miss AGN in low-metallicity hosts (see discussion in Section 5 in Reines et al. 2013). Additionally, as the referee points out, AGN can be obscured by dust in the optical. We nevertheless conclude that spectra of all point sources in a LIGO area of 50 deg^2 , which is only 0.5% of the SDSS footprint, and would require a comparable sensitivity, could feasibly be obtained by a follow-up campaign.

Finally, we also believe such a follow-up could be feasible in X-rays. The Wide Field Imager (WFI) instrument on the proposed Athena mission (www.cosmos.esa.int/web/athena) has a field of view of 0.5 deg^2 and an effective area of $\sim 1 \text{ m}^2$ at 1 keV. Consider a $M=10^6 M_{\text{sun}}$ BH at a distance of $d=0.7$ Gpc, emitting $L_x=0.03 L_{\text{edd}}$ at 1 keV. We get a flux $F_x=$

$0.03 \times 10^6 \times 1.3 \times 10^{38} / (4 \times 3.14 \times 3 \times 10^{27} \times 3 \times 10^{27} \times 0.7^2) = 0.8 \times 10^{-13} \text{ erg/s/cm}^2$. A 1 ksec integration over the $1 \text{ m}^2 = 10^4 \text{ cm}^2$ area then gives $N = 1000 \times 10^4 \times 0.8 \times 10^{-13} / 1.6 \times 10^{-9} = 500$ counts, i.e. a very significant detection of the X-ray point source. Covering a total area of 50 deg^2 will require a tiling with 100 such pointings, with a total 100 ksec; an exposure, or ~ 30 hours.

COMMENT: 2-study the feasibility of the study accounting for possible incompleteness of Quasar catalogs. For example, how does the performance of the method scales if a fraction x of the BHBs are in fact hosted by quasars, but those are not identified in companion surveys?

RESPONSE: We added specification of the dependence of the results from the completeness of host catalogs to Figure 1, see x axis label and caption.

COMMENT: I think addressing point 2 is essential for drawing sensible astrophysical conclusions. If a fraction x of AGNs are missing (because obscured or below the threshold of optical surveys) and a fraction y of the LIGO BHB comes from AGN, what would this technique tell you? I would assume there might be degeneracies between the 2 parameters (x and y), can that be confidently broken?

RESPONSE: x and y are indeed degenerate. There are two relevant questions to this problem. The first one, which we aim to address in the manuscript, is whether we can demonstrate that some of the BBH mergers originate from AGN. Answering this question is more difficult with incomplete catalogs, but is otherwise not affected. With sufficient number of measurements, the question can be answered. Our Fig 1 specifies the effect of incomplete catalogs on how many measurements are needed for this. The second question is whether we can determine the specific fraction of BBH mergers that originate from the AGN channel (that is, not just that there are such, but the actual fraction). To answer this question (which we don't discuss in the manuscript), one needs to know the incompleteness of the catalog to resolve the x - y degeneracy.

To clarify this point to the reader, we added an extra point paragraph 8 of the Discussion section, which states that in order to establish the actual fraction of AGN-BBH, one needs to have knowledge of the level of completeness of the AGN catalog in the GW localization volumes.

COMMENT: a-After the introduction, the Authors present the results right away. I am not sure this is required in the Nature Communications format. In any case, a bit more of background should be given before this discussion. For example, neither N_{gw} nor f_{agn} are defined at this point, so that equation 1 is essentially meaningless at first glance (well, one can guess, of course). Please define the quantities and spend few lines to frame the results.

RESPONSE: We moved and modified three paragraphs from the beginning of Methods to the beginning of Results to make the transition between the introduction and results more continuous and to ensure that the reader follows the description, and has all the parameters defined, even without needing to jump to the Methods section. We further added a short paragraph to the beginning of the introduction to make the transition between the abstract and the introduction smoother.

COMMENT: b-When talking about the applicability of this technique to other formation channels, the Authors write:

"The number of globular clusters within a galaxy strongly correlates with the mass of the central supermassive black hole , so relatively rare, large black holes near the turnover in the Schechter function are the dominant contributors."

One may argue that this correlation is just a mirror of the SMBH-bulge mass relation and of the fact that the number of globular clusters in spheroidal galaxies is proportional (quite unsurprisingly) to the mass of the spheroid. I do not see how this is different than claiming that the BHBs are most likely originating from galaxies with more stars. Unless the Authors can demonstrate this is not the case, this sentence should be removed.

RESPONSE: The Referee raises a valid point. We removed the sentence.

Reviewer #2 (Remarks to the Author):

COMMENT: The authors derived the condition that the black hole merger happens at the AGNs at a statistically significant level. Even if the future observations would prove it, the statistical analysis cannot say exactly where the merger occurs. Also why the merger prefers to happen at AGNs still remains mystery. The key to solve these questions is the observations of the electromagnetic counterparts. Through them, one can know where and how the binary stellar-mass black holes are formed, and evolve towards the coalescence. That is why this work should not be so impressive for the general readers.

RESPONSE: We agree with the Referee that the observation of electromagnetic counterparts would be the best way to probe the BBH host and formation channel. Unfortunately, we do not typically expect any observable electromagnetic emission from BBH mergers. While it is plausible that in some cases counterparts may be detected (see the authors work on some possibilities in [1602.03831] and [1602.04226], as well as DeMink & King [arXiv:1703.07794v1], Murase et al. [arXiv:1602.06938] and Perna et al. [arXiv:1602.05140]), current predictions have large uncertainties and the current general consensus is that most cases will produce no counterparts. This is the motivation of our present work, which provides an alternative probe of the formation channel even in the likely scenario in which electromagnetic observations will not be available.

COMMENT: It would be better for the readers to make the following clearer; the concrete values of the significant level and the significance probability, and their derivation basis. This is because whether the null hypothesis can be rejected clearly depends on their values. If it cannot be rejected, the authors cannot say anything conclusive statistically.

RESPONSE: We added text to the beginning of the Results section (see in particular paragraph 4) that introduce Ngw3sigma. We also point the reader here to the Methods section, in which a more detailed, precise definition and its derivation can be found. We use 3sigma significance probabilities in this work, more precisely a median 3sigma, which means that after Ngw3sigma observations, we will have a 50% chance of being able to constrain the host at or more than 3sigma level.

Reviewer #3 (Remarks to the Author):

COMMENT: The authors make an attempt to justify that association of gravitational wave sources with rare host galaxies may be identified statistically.

It is interesting that a very similar case was discussed in paper by Burenin et al. (<http://adsabs.harvard.edu/abs/1998AstL...24..427B>, arXiv:astro-ph/9804274). In this work the authors discovered the association of gamma-ray bursts with AGNs with almost exactly the same method, which is developed in the paper by Bartos et al. Note, that the number of objects considered and the typical size of their localizations in the sky are also very similar.

The association of bright GRBs with AGNs at $z \sim 0.3$ was found at more than 3 sigma confidence by Burenin et al., but as we know today GRB are not associated with AGN directly. The reason of observed association is that GRB hosts follow the same large scale structure as other objects at the same distances and the correlation length (~ 6 Mpc, i.e. 0.4 deg at $z \sim 0.3$) is in fact of the same order as the size of GRB localization region (1-2 deg). Therefore, the result is correct, i.e. that was indeed one of the first confirmation that the GRB hosts are located at cosmological distances. However, I think that this example shows very clearly that even if the association between the populations is found with this method, that does not necessarily mean that the populations are directly related.

RESPONSE: We thank the Referee for bringing this paper to our attention. We agree it is very relevant to our work. However, we believe that in our context, the AGN-galaxy cross-correlation is a much smaller effect. For the best-localized events we rely on, the error volume is approximately 10^5 Mpc^3 . Assuming this is a sphere, the radius of the sphere is $R = [3 \times 10^5 \text{ Mpc}^3 / (4 \times \pi)]^{1/3} = 29 \text{ Mpc}$. This is 4.5 times the cross-correlation length, $r_0 = 6.6 \text{ Mpc}$ between AGN and galaxies in Shen et al. 2013. This implies that if the LIGO event occurred in a random galaxy, then the probability of finding a quasar in the LIGO volume, in excess of random, in a $R = 29 \text{ Mpc}$ sphere around this galaxy, would be $6.6 \times (29/6.6)^{-1.69} = 0.08$. (Using the slope 1.69 of the cross-correlation function from Shen et al. 2013.) This produces only a much weaker LIGO event - Quasar cross-correlation. Thus, we agree that the GRB-QSO analysis in Burenin+1998 is intriguingly similar to our proposal, but in that case, the error volume is close to the correlation length which could explain why they found a 'fake' cross-correlation with quasars via galaxies. In our case, the error boxes are ~ 5 times larger than the (three-dimensional) correlation length, and this effect is much smaller.

COMMENT: In the case of LIGO/Virgo GW events, the localization area of well localized ones should be < 10 square degrees (Chen & Holz, 2016), i.e. their typical sizes should be smaller than ~ 3 deg. It is larger, but still of order of the correlation angular scale of galaxies at $z \sim 0.1-0.3$ (0.3--1 deg). That is why I do not agree with the assumption made by authors at page 6, that the effect of cross-correlation of galaxies with quasars at this angular scale should be small. This is one of the main assumption of the method, so the results may change very significantly if it would be relaxed.

RESPONSE: In our context, AGN-galaxy cross-correlation will be the relevant quantity. As we describe in the previous point, the AGN-galaxy correlation length is several times smaller than the resolution of the typical best reconstructed GW events (for localization volume 10^5 Mpc^3),

therefore we expect this to be a small effect. Cross correlation can become a problem once the localization volume improves to 10^3 Mpc^3 , at which size the correlation length and the localization error are the same. This precision, however, is essentially unreachable with second generation GW instruments, for binary black hole mergers at least.

Reviewer #1 (Remarks to the Author):

The Authors carefully considered all the criticism and made a good job in stressing better what are the requirements and the caveats related to the study.

The point related to targeted follow-ups though needs further clarification, because I think it is invalid, as stated now. It is not sufficient to follow up the volumes of the identified LIGO sources. In absence of a control sample of random fields probed to the same depth in fact, the statistics of AGNs in the LIGO event fields cannot be compared to any null-hypothesis. One might well find that each individual field contains a faint AGN, but no conclusion can be drawn from there, if the statistics of faint AGNs in a 'control sample' of fields unrelated to LIGO events is taken. Unless I misinterpreted the Author point, this can be further clarified.

Other than this, I am happy with the current version of the manuscript.

Reviewer #3 (Remarks to the Author):

As it is written in their response, the authors agree that the number of AGNs near any galaxy will be $\sim 10\%$ in excess to random in typical GW error volume. In that case, if order of 1000 GW events are used in their study, the authors will detect GW-AGNs association at near 3 sigma level even if GW events are in fact associated with all galaxies, not with AGNs specifically.

If this effect is taken in account, it will change significantly the curves at Figure 1 for smaller f_{agn} and for larger sources densities. That is what I was trying to note in my former report. I am sorry, if it was not explained very clearly.

Again, I think that AGN--galaxy cross-correlation should be taken in account in this study.

We thank our Referees for the useful additional feedback. We modified the description to clarify our suggested treatment of follow-up galaxy surveys as our first Referee suggests, and numerically investigated the effect of correlations on our results in order to address our third Referee's concern. Please find details below, changes in the manuscript are marked in red. We hope that the updated manuscript is now ready for publication in Nature Communications.

Reviewer #1 (Remarks to the Author):

COMMENT: The point related to targeted follow-ups though needs further clarification, because I think it is invalid, as stated now. It is not sufficient to follow up the volumes of the identified LIGO sources. In absence of a control sample of random fields probed to the same depth in fact, the statistics of AGNs in the LIGO event fields cannot be compared to any null-hypothesis. One might well find that each individual field contains a faint AGN, but no conclusion can be drawn from there, if the statistics of faint AGNs in a 'control sample' of fields unrelated to LIGO events is taken. Unless I misinterpreted the Author point, this can be further clarified.

RESPONSE: We thank the Referee for this clarification, indeed our suggested method of on-the-fly surveys require a substantial number of control samples. We added text to the very end of the Discussion section to point out that follow-up surveys are only feasible if these control samples are available.

Reviewer #3 (Remarks to the Author):

COMMENT: As it is written in their response, the authors agree that the number of AGNs near any galaxy will be ~10% in excess to random in typical GW error volume. In that case, if order of 1000 GW events are used in their study, the authors will detect GW-AGNs association at near 3 sigma level even if GW events are in fact associated with all galaxies, not with AGNs specifically. [...] If this effect is taken in account, it will change significantly the curves at Figure 1 for smaller f_{agn} and for larger sources densities. That is what I was trying to note in my former report. I am sorry, if it was not explained very clearly.

RESPONSE: To quantitatively investigate the role of AGN-galaxy correlations, we considered the corresponding cross-correlation function from Shen et al 2013. According to their results, we take the cross-correlation function to be $\xi(r)=(r/6.6h^{-1} \text{ Mpc})^{-1.69}$. The number of AGN expected in the GW error volume for our null hypothesis is

$$\lambda_i = \rho_{\text{agn}} * V_i * (1 + \xi(r_i))$$

With the rightmost term being the new term due to the correlations. Here,

$$x_i = (1/V_i) * \int_0^{r_i} \xi(r) * 4\pi * r^2 dr$$

is the integrated excess due to correlation within the error volume, with $r_i = (3V_i/4\pi)^{1/3}$ being the radius corresponding to V_i , where for simplicity we assume that the GW error volume is spherical.

We used this new average to calculate Eqs. 2 and 3, leaving everything else the same in our analysis, and with this we reproduced Fig. 1. We find no visible difference from the results obtained without taking into account correlations.

To cross-check this result, we numerically calculated x_i to get an idea of how many additional AGN are expected within the relevant volume due to correlations. We find that at $10^{4.75}$ Mpc³ volume, within which we expect 1 AGN, x_i is ~ 0.5 , showing that the excess AGN due to correlations is sub-dominant.

To inform the reader of these results, we added a paragraph to the Discussion in which we describe that we repeated our calculations by additionally taking into account galaxy-AGN correlations, but find no difference in the final results, concluding that our initial assumptions were reasonable.

Reviewer #1 (Remarks to the Author):

I am happy with the revision proposed by the Authors.

Reviewer #3 (Remarks to the Author):

Ok, I think that now the paper can be recommended for publication.